# Glycated albumin in the detection of diabetes during COVID-19 hospitalization

**Fernando Chimela Chume**[1,2,3], **Priscila Aparecida Correa Freitas**[3,4], **Luisa Gazzi Schiavenin**[3], **Eduarda Sgarioni**[5], **Cristiane Bauermann Leitao**[1,3,6], **Joíza Lins Camargo**[1,3,5]*

1 Post-Graduate Program in Medical Sciences: Endocrinology, Universidade Federal do Rio Grande do Sul, Porto Alegre, Brazil, 2 Faculty of Health Sciences, Universidade Zambeze, Beira, Mozambique, 3 Diabetes and Metabolism Group, Centro de Pesquisa Clínica, Hospital de Clínicas de Porto Alegre, Porto Alegre, Brazil, 4 Laboratory Diagnosis Division, Clinical Biochemistry Unit, Hospital de Clínicas de Porto Alegre (HCPA), Porto Alegre, Brazil, 5 Experimental Research Centre, Hospital de Clínicas de Porto Alegre, Porto Alegre, Brazil, 6 Endocrinology Division, Hospital de Clínicas de Porto Alegre (HCPA), Porto Alegre, Brazil

* jcamargo@hcpa.edu.br

**Data Availability Statement:** All relevant data are within the paper and its Supporting Information files.

## Abstract

### Background

Diabetes has emerged as an important risk factor for COVID-19 adverse outcomes during hospitalization. We investigated whether the measurement of glycated albumin (GA) may be useful in detecting newly diagnosed diabetes during COVID-19 hospitalization.

### Methods

In this cross-sectional test accuracy study we evaluated HCPA Biobank data and samples from consecutive in-patients, from 30 March 2020 to 20 December 2020. ROC curves were used to analyse the performance of GA to detect newly diagnosed diabetes (patients without a previous diagnosis of diabetes and admission HbA1c $\geq$6.5%).

### Results

A total of 184 adults (age 58.6 ± 16.6years) were enrolled, including 31 with newly diagnosed diabetes. GA presented AUCs of 0.739 (95% CI 0.642–0.948) to detect newly diagnosed diabetes. The GA cut-offs of 19.0% was adequate to identify newly diagnosed diabetes with high specificity (85.0%) but low sensitivity (48.4%).

### Conclusions

GA showed good performance to identify newly diagnosed diabetes and may be useful for identifying adults with the condition in COVID-19-related hospitalization.

**Funding:** This work was supported by the Research Incentive Fund (FIPE) of the Hospital de Clinicas de Porto Alegre (HCPA) (FIPE/HCPA, GPPG 2021-0256), and by the Brazilian National Council for Scientific and Technological Development (CNPq 401610/2020-9, Chamada MCTIC/CNPq/FNDCT/MS/SCTIE/Decit N 07/2020). FCC received scholarship from Programa de Excelência Acadêmica da Coordenação de Aperfeiçoamento de Pessoal de Nível Superior – Finance Code 001 (CAPES-PROEX). LGS received an undergraduate scholarship from Fundação de Amparo à Pesquisa do Estado do Rio Grande do Sul (FAPERGS). The funders had no role in study design, data collection and analysis, decision to publish, or preparation of the manuscript.

**Competing interests:** The authors have declared that no competing interests exist.

**Abbreviations:** AUC, Areas under the curve; COVID-19, coronavirus disease 2019; DCCT, Diabetes Control and Complications Trial; eGFR, estimated glomerular filtration rate; GA, Glycated albumin; GSP, Glycated Serum Protein; HCPA, Hospital de Clinicas de Porto Alegre; HPLC, high performance liquid chromatography; K2EDTA, dipotassium ethylenediaminetetraacetic acid; LR, likelihood ratio; NGSP, National Glycohemoglobin Standardization Program; NPV, negative predictive value; PPV, positive predictive value; RBG, random blood glucose; ROC, Receiver Operating Characteristic; RT-PCR, real-time reverse-transcriptase–polymerase-chain-reaction; SARS-CoV-2, severe acute respiratory syndrome coronavirus 2; SD, standard deviation; STARD, Standard for Reporting Diagnostic Accuracy; USA, United States of America; 24 h-UPE, 24-h urinary protein excretion.

## Introduction

Diabetes and hyperglycaemia *per se* have emerged as important risk factors for hospitalization, acute respiratory distress syndrome, and death in patients with coronavirus disease 2019 (COVID-19) [1–4]. These findings represent a worldwide public health problem considering the actual diabetes burden. Besides, the diagnosis of diabetes is neglected and estimative point to over half of adults living with diabetes are undiagnosed [5].

Many reasons may explain the susceptibility of people with diabetes and/or uncontrolled hyperglycaemia to develop adverse events following COVID-19 infection, including the direct effect of hyperglycaemia in the immune system [6–8]. It is assumed that hyperglycaemia at the time of hospital admission increases the risk of poor outcomes, regardless of prior diabetes status, and that the achievement of glycaemic targets, soon after admission, may significantly improve prognosis [8,9]. Although this background suggests the importance of assessing glycaemic status on admission, the significance of blood glucose levels, by glucose-based test and/or HbA1c, at the time of admission for the management of COVID-19 people remain unclear. In general, studies have reported significant association between blood glucose and/or HbA1c on admission with COVID-19 adverse outcomes [9–15]; while some studies showed no associations [9,10,16–18].

Current strategies for glycaemic parameters evaluation in hospitalized patients have limitations. Blood glucose, widely used to identify and control hyperglycaemia in hospitalized patients, provides one point of blood glucose, which may be affected by fasting, food intake and acute illness such as COVID-19 [7,13], and is susceptible to pre-analytical interferences [19,20]. On the other hand, HbA1c overcomes these issues since it presents few pre-analytical interference and low intra-individual variability [21]. Furthermore, HbA1c results are not affected by fasting and acute illness. It is well established that an admission HbA1c $\geq 6.5\%$ suggests diabetes previous to hospitalization [22]. Nevertheless, HbA1c has its own limitations, since its results are not accurate in conditions with altered erythrocyte turnover, such as recent transfusion, blood loss, anaemia, and chronic kidney disease [23]. In fact, anaemia frequently emerges in people with COVID-19 wherein HbA1c values may not accurately reflect blood glucose concentrations [24]. In view of this scenario, it is important to consider alternative options of glycaemic markers in hospitalized patients.

Glycated albumin (GA) is a test that has gained prominence as an alternative glycaemic marker. GA reflects short-term mean glycaemia (2–3 weeks), rather than 2–3 months observed for HbA1c [25]. Unlike HbA1c, GA is haemoglobin/erythrocyte independent, but their performance in the diagnosis of diabetes in the general population is similar [26–30]. Therefore, it is advocated that GA is a useful alternative to HbA1c under conditions where the latter does not reflect glycaemic status precisely. In addition, increased GA levels have been shown to predict the onset of microvascular and macrovascular outcomes, and even death [31–33]. An evaluation of multiple glycaemic markers (blood glucose, HbA1c, GA and GA/HbA1c ratio) performed at the time of admission [18] showed that only GA and GA/HbA1c ratio predicted progression of COVID-19 from mild to severe disease in people with type 2 diabetes. However, no study evaluated the performance of GA as a predictor of glycaemic status in hospitalized patients with COVID-19.

Therefore, this study was designed to evaluate the performance of GA to identify patients presenting newly diagnosed diabetes on hospital admission.

## Materials and methods

### Study design

This is a retrospective study designed to use data and samples stored in the Hospital de Clinicas de Porto Alegre (HCPA) Biobank [34]. All participants provided written informed consent

to take part in the HCPA Biobank. This study was reviewed and approved by the Research Ethics Committee of HCPA (GPPG 2021–0256) and by HCPA Biobank. We reported this study of diagnostic accuracy according to Standard for Reporting Diagnostic Accuracy (STARD) initiative guidelines [35].

## Participants and data collection

Consecutive patients admitted at HCPA Emergency Department between 30 March 2020 and 20 December 2020 due to COVID-19 infection were enrolled. Study inclusion criteria were adults (>18 years old) without history of diagnosed diabetes with admission blood samples stored in the HCPA Biobank and a laboratory confirmation (by real-time reverse-transcriptase–polymerase-chain-reaction, RT-PCR) of severe acute respiratory syndrome coronavirus 2 (SARS-CoV-2) infection.

Exclusion criteria were comprised factors that potentially could affect HbA1c or GA results: severe hypoalbuminemia (<3 g/dL), anemia (haemoglobin <7 g/dL) or blood transfusion at admission, chronic kidney disease with estimated glomerular filtration rate (eGFR) at admission of $\leq$ 15 ml/min/1.73m$^2$, presence of variant haemoglobin, dialysis, history of hepatic cirrhosis, documented nephrotic syndrome [24-h urinary protein excretion (24 h-UPE) >3.0 g/day and/or serum albumin <3.0 g/dl], active rheumatic disorder and/or untreated thyroid dysfunction. We also excluded patient hospitalized for less than 24 h, pregnant women, individuals receiving anti-diabetic drugs or with history of chronic use of immunosuppressant and renal transplant recipients.

From 4 March to 20 May 2022, data were collected using a query to the HCPA Covid-19 Biobank Data [36]. In case of missing relevant data, two authors independently attempted to extract the information from medical records (missing data was last updated on 27 January 2023). If admission blood glucose and HbA1c data were not performed as part of usual patient care, they were measured in stored Biobank aliquot samples.

## Definitions

Hospitalization was defined as patient presence in the hospital for more than 24 h. Diagnosis of COVID-19 illness was defined as a positive SARS-CoV-2 result detected by RT-PCR assay in a specimen collected on a nasopharyngeal swab. Presence of diabetes was defined by history of diabetes, prescription for insulin and/or an oral antidiabetic medication in use prior COVID-19 admission. Newly diagnosed diabetes was defined as HbA1c $\geq$6.5% in individuals without established diagnosis of diabetes prior COVID-19 admission. Admission lab tests were defined as those in samples collected within 24 hours of hospital presentation.

## Laboratory analysis

All analyses were carried out in blood samples collected at the beginning of hospitalization for assistance purposes and stored at HCPA Biobank. Random blood samples were drawn into Vacutainer® tubes (BD, New Jersey, USA) with gel and without anticoagulant for serum samples and with K2EDTA for whole blood samples. Serum separated immediately after centrifugation and whole blood were stored in aliquots at −80˚C freezer until analysis that were performed on 12 October 2022.

**Admission Random Blood Glucose (RBG) analysis.** Glucose was measured in random samples collected at the time of admission by an enzymatic method in the biochemistry automated analyzer Alinity C (Abbott Laboratories, Illinois, USA).

**HbA1c analysis.** HbA1c was measured in whole blood by high performance liquid chromatography (HPLC) using VARIANT II™ System (BioRad Laboratories, Hercules, CA, USA).

This assay is certified by the National Glycohemoglobin Standardization Program (NGSP) and aligned to the DCCT reference and the International Federation of Clinical Chemistry reference (http://www.ngsp.org/ifcc.asp). The stability of the HbA1c assay in long-term stored specimens has already been evaluated [21].

**Glycated Albumin (GA) analysis.** Glycated Serum Protein (GSP) was determined by an enzymatic method (GlycoGap®, Diazyme Laboratories, Poway, CA). Total albumin was measured with bromocresol green colorimetric method. Both GSP and albumin were achieved by the automated analyzer Alinity C (Abbott Laboratories, Illinois, USA). As GlycoGap® assay quantifies the total of GSP (μmol/L), the results are converted to percent of GA by the following equation provided by the manufacturer: GA (%) = {[GSP (μmol/L) x 0.182 + 1.97]/total albumin (g/dL)} + 2.9 [25].

## Statistical analysis

Continuous variables were expressed as mean and standard deviation (SD) for normally distributed variables and as median (interquartile range) for non-Gaussian variables. Data normality was examined using histograms and the Shapiro-Wilk test. Student's T test or Mann-Whitney U test were used for continuous variables when appropriate. Categorical variables were expressed as numbers and frequencies (%) and the Chi-square or Fisher's exact test were used to examine the significance of the contingency. Receiver Operating Characteristic (ROC) curve was used to analyse the performance of GA to detect newly diagnosed diabetes. The sensitivity, specificity, likelihood ratios (LR), positive predictive value (PPV), and negative predictive value (NPV) were calculated for different GA cut-offs. Also, the optimal cut-off for GA was derived from the ROC curve with the shortest distance to sensitivity and specificity by the Youden index (Y = sensitivity + specificity– 1). The GA first cut-off with specificity >0.85 was chosen as the diagnostic criterion point. Venn diagram was used to present the number of individuals identified by each test (GA and HbA1c) and their overlaps. For clinical applicability, we presented the post-test probabilities using the Fagan's Nomogram [37]. Based on literature data, pre-test probability for newly diagnosed diabetes was considered 19% [38].

The IBM SPSS software for Windows, version 20.0 (Statistical Package for Social Sciences—Professional Statistics, IBM Corp, Armonk, USA) and MedCalc, version 19.1 (MedCalc software, Ostend, Belgium) were used for data analysis. P values 0.05 were considered significant.

## Results

We identified 212 potentially eligible individuals with COVID-19 infection requiring hospital admission at HCPA Emergency Department between 30 March 2020 and 20 December 2020. Of those, 184 participants were enrolled in the present study (S1 Table). The flowchart with reasons for exclusions is presented in Fig 1.

The clinical and laboratory characteristics of all individuals are shown in Table 1. The mean age of the participants was 58.6 ± 16.6years, 50.5% were women, 78.8% were Caucasian, and median admission BMI was 29.4 (25.7, 34.5) kg/m$^2$. The length of hospital stay was 9 days (interquartile range of 4 and 16 days). GA, HbA1c and RBG values were not normally distributed, and their medians (interquartile range) were 16.4% (14.9%, 18.3%), 5.7% (5.3%, 6.2%) and 113.0 mg/dL (97.0 mg/dL, 140.0 mg/dL), respectively. Thirty-one (16.8%) participants were newly diagnosed with diabetes after admission for COVID-19. Hypertension was the most common comorbidity with a prevalence of 42.9%.

The AUC for GA in the detection of newly diagnosed diabetes by HbA1c ≥6.5% was good with an AUC of 0.739 (95% CI 0.642–0.948) (Fig 2).

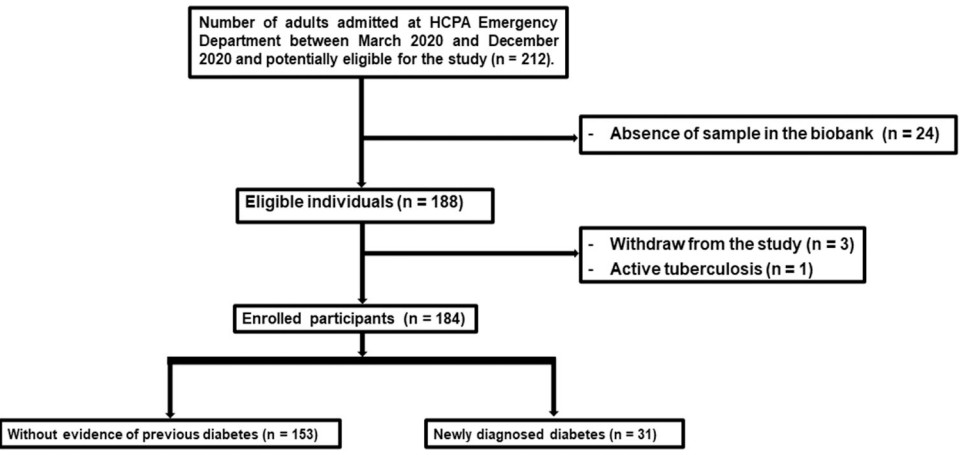

**Fig 1. Flowchart with reasons for exclusions from the study.**

The optimal cut-off value for GA was 17.5% with sensitivity of 74.2%, and specificity of 73.9%. This cut-off also presented the maximum value of the Youden index and the LR+ and LR- were 2.8 and 0.3, respectively. GA of 19.0% presented greater specificity (85.6%), although lower sensitivity (48.4%) (Table 2). GA ≥19.0% correctly identified 146 individuals (15 true positives and 131 true negatives) and misclassified 16 individuals with newly diagnosed diabetes by HbA1c ≥6.5%. Twenty-three individuals had GA ≥19.0% without diabetes by HbA1c ≥ 6.5% (Fig 3). However, 14 of them had in-hospital hyperglycemia requiring insulin therapy prescription and/or HbA1c levels within the range for pre-diabetes (5.7 to 6.4%) (result not shown). Besides, using the Fagan's nomogram (Fig 4), we estimated that after a positive test (GA≥19.0%) the post-test probability for diabetes would increase to 43%, while after a negative test (GA<19.0%) would decrease to 13% (Fig 4).

## Discussion

Our results showed that GA showed a good performance to detect newly diagnosed diabetes in hospitalized individuals with COVID-19.

As far as we know, this is the first study that evaluated the diagnostic accuracy of GA to detect diabetes in hospitalized individuals with COVID-19. However, our findings are supported by studies conducted in the general population that demonstrated very good GA performance for diabetes diagnosis [26–30]. In a systematic review and meta-analysis of diagnostic test accuracy, we showed that GA performed very well for diabetes diagnosis by oral glucose tolerance test with/without HbA1c in the non-hospitalized general population [26]. One community-based study of non-hospitalized Japanese adults reported that GA had excellent ability to identify diabetes defined by FPG or HbA1c [28]. A multi-ethnic community-based study in non-institutionalized American adults also showed that GA had very good ability to identify diabetes defined by FPG; HbA1c; FPG or HbA1c; FPG and HbA1c, with high AUC to all definitions [29]. In another community-based study was reported good performance of GA for detection of diabetes by FPG or HbA1c [30]. As the first study of diagnostic accuracy of GA in hospitalized individuals, our study is in agreement with findings in the literature by showing the clinical utility of GA also in the hospitalization admission.

The GA cut-off of 19.0% was useful to detect new cases defined by HbA1c ≥6.5% as reference. This cut-off has good specificity and NPV, but low sensitivity and PPV. Considering the LR+ of 3.2, it is possible to assume that patients with newly diagnosed diabetes were about 3

**Table 1. Clinical and laboratory characteristics of participants in the study.**

| Characteristics | All participants (n = 184) |
|---|---|
| Age (years) | 58.6 ± 16.6 |
| Gender [female n (%)] | 93 (50.5) |
| Ethnicity | |
| Caucasian [n (%)] | 145 (78.8) |
| sub-Saharan African [n (%)] | 34 (18.5) |
| Multi or other ancestry [n (%)] | 5 (2.7) |
| Length of hospital stay (days) | 9 (4, 16) |
| In-hospital hyperglycaemia requiring insulin therapy prescription [n (%)] | 112 (60.9%) |
| On admission | |
| BMI (kg/m$^2$) | 29.4 (25.7, 34.5) |
| GA (%) | 16.4 (14.9, 18.3) |
| HbA1c (%) | 5.7 (5.3, 6.2) |
| HbA1c ≥6.5% [n (%)] | 31 (16.8) |
| HbA1c ≥7% [n (%)] | 16 (8.7) |
| Random blood glucose (mg/dL) | 113.0 (97.0, 140.0) |
| GA/HbA1c ratio | 2.9 ± 0.6 |
| Serum albumin (mg/dL) | 3.7 (3.4, 4.0) |
| Haemoglobin (g/dL) | 13.2 ± 1.7 |
| Hematocrit (%) | 39.4 ± 4.7 |
| White blood cells (x10$^3$/μL) | 7.330 (5.310, 9.755) |
| Platelets (x10$^3$/μL) | 227.0 (170.0, 289.0) |
| C-reactive protein (mg/dL) | 93.1 (41.9, 158.8) |
| Serum creatinine (mg/dL) | 0.87 (0.73, 1.06) |
| eGFR (mL/min/1.73 m2) | 85.0 (66.0, 94.0) |
| Urea (mg/dL) | 35.0 (24.3, 46.0) |
| Newly diagnosed diabetes [n (%)] | 31 (16.8) |
| History | |
| Hypertension [n (%)] | 79 (42.9) |
| Ischemic heart disease [n (%)] | 11 (6.0) |
| Stroke [n (%)] | 8 (4.3) |

Data are expressed as mean ± SD, median (interquartile range) or frequencies; eGFR, estimated glomerular filtration rate by CKD-EPI Creatinine Equation; GA, glycated albumin; HbA1c, glycated haemoglobin; SpO2, saturation of partial pressure oxygen; Newly diagnosed diabetes was defined as HbA1c ≥6.5% in patients without established diagnosis of diabetes previous the COVID-19 admission.

times more likely to have a GA ≥19.0%. The use of this cut-point alone would fail to diagnose 16 positive cases and 23 were false positive in our study. Despite the false-negative effects due to low sensitivity for screening, the use of GA ≥19.0% would act as an alternative or additional tool to identify individuals at risk for diabetes, since 14 out of the 23 individuals with GA ≥19.0% but without HbA1c ≥6.5%, had in-hospital hyperglycemia requiring insulin therapy prescription and/or HbA1c levels within the range for pre-diabetes. In addition, the post-test probability for diabetes was 43%, greater than two times the pre-test probability (19%). Newly diagnosed diabetes by HbA1c ≥6.5% has been associated with increased risk for adverse outcomes and mortality in COVID-19 patients [38,39]. There is no data about the association between GA levels and the development of COVID-19 adverse outcomes in individuals without a previous diagnosis of diabetes. However, in the general population, increased GA, with

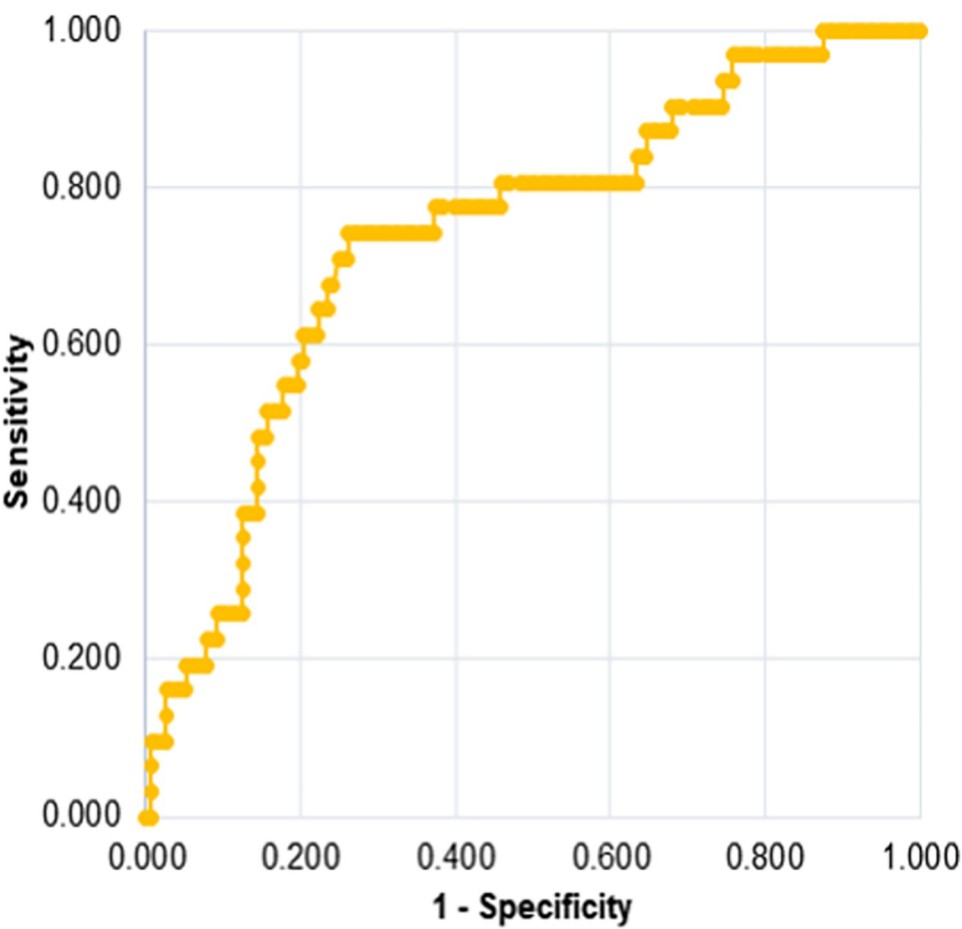

**Fig 2. Receiver operating characteristic (ROC) curve to access the performance of admission GA to detect newly diagnosed diabetes by HbA1c ≥6.5% (n = 184).** AUC, area under the curve; CI, confidence interval; HbA1c, glycated haemoglobin; GA, glycated albumin; SE, standard error.

predictive values similar to HbA1c, has been shown to predict the onset of microvascular and macrovascular outcomes, and death [31–33]. The risk of the outcomes and death starts in the prediabetes stage even before clinical diabetes sets in [31–33]. This behavior is explained by the fact that there is no reference standard definition with nearly perfect sensitivity and specificity for detecting diabetes and the risk of its complications. Consequently, all tests are equally appropriate to diagnose diabetes. In our study, the adequate cut-off to detect newly diagnosed diabetes was slightly higher than those reported in general population, where GA ranged between 15% to 18% to identify diabetes [26–30]. We did not explore time to admission from onset of COVID-19-related symptoms for this study.

Nevertheless, these results may suggest that, in general, acute illness such as COVID-19 may slight alter the interpretation of GA when assessing glycaemic status, but there are no studies on the subject. We understand that this topic is clinically relevant, and properly designed studies are needed to elucidate why, and how COVID-19 may interfere with GA measurements. We believe that GA might be useful in other acute infectious diseases with similar issue, but further studies are necessary for accurate conclusions about this issue.

**Table 2. Performance of different cut-offs of GA to detect newly diagnosed diabetes.**

| Index Test | Cut-point | Newly diagnosed diabetes. (N = 184; prevalence = 16.7%) | | | | | |
|---|---|---|---|---|---|---|---|
| | | Sensitivity (%) | Specificity (%) | LR+ | LR- | PPV | NPV |
| GA (%) | 15.0 | 90.3 | 28.1 | 1.3 | 0.3 | 0.903 | 0.281 |
| | 15.5 | 83.9 | 35.3 | 1.3 | 0.5 | 0.839 | 0.353 |
| | 16.0 | 80.6 | 45.8 | 1.5 | 0.4 | 0.806 | 0.458 |
| | 16.5 | 77.4 | 56.2 | 1.8 | 0.4 | 0.774 | 0.562 |
| | 17.0 | 74.2 | 64.7 | 2.1 | 0.4 | 0.742 | 0.647 |
| | 17.5 | 74.2 | 73.9 | 2.8 | 0.3 | 0.742 | 0.739 |
| | 18.0 | 61.3 | 78.4 | 2.8 | 0.5 | 0.613 | 0.778 |
| | 18.5 | 51.6 | 82.4 | 2.9 | 0.6 | 0.516 | 0.824 |
| | 19.0 | 48.4 | 85.0 | 3.2 | 0.6 | 0.484 | 0.850 |
| | 19.5 | 38.7 | 85.6 | 2.7 | 0.7 | 0.387 | 0.856 |
| | 20.0 | 25.8 | 87.6 | 2.1 | 0.8 | 0.258 | 0.876 |
| | 20.5 | 25.8 | 90.2 | 2.6 | 0.8 | 0.258 | 0.902 |
| | 21.0 | 19.4 | 94.1 | 2.9 | 0.8 | 0.226 | 0.922 |
| | 21.5 | 19.4 | 94.1 | 3.3 | 0.9 | 0.194 | 0.941 |
| | 21.7 | 19.4 | 94.8 | 3.7 | 0.9 | 0.194 | 0.948 |

GA, glycated albumin.

Our study had several strengths. It is the first to assess the performance of GA on admission of COVID-19 hospitalized adults. We attempted to remove confounding factors by excluding individuals with known interfering factors for GA and HbA1c and we followed the STARD 2015 reporting guideline for diagnostic accuracy studies [35] to assure reporting the results adequately.

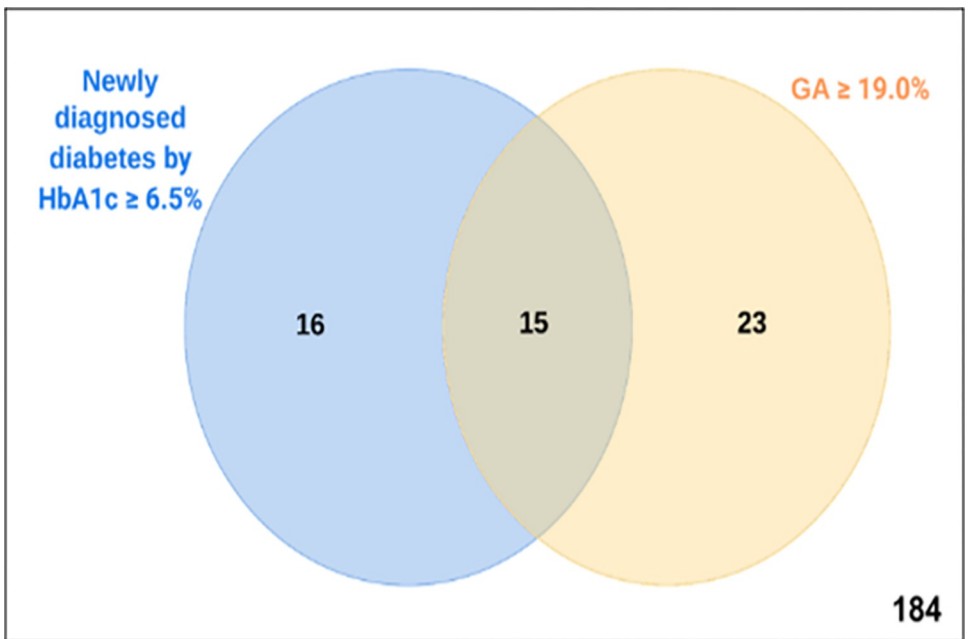

**Fig 3. Number of individuals identified by each test (GA and HbA1c) and overlaps.** HbA1c, glycated haemoglobin; GA, glycated albumin.

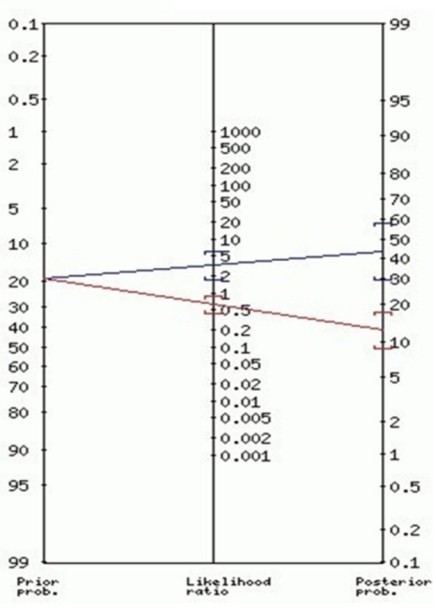

**Fig 4. Fagan's nomogram for GA shows pre- and post-test probabilities for newly diagnosed diabetes.** GA, glycated albumin.

There were also some limitations to our study. First, the sample size is small, but it was calculated a priori to assure the study power of 80% and an estimated alfa error of 5%. Second, it was not possible to perform oral glucose tolerance tests or fasting blood glucose, as this research was carried out with data and samples from a Biobank. However, we relied on admission HbA1c for the reference test, a marker recommended as suitable for the study setting [22]. Third, due to cross-sectional design, GA and HbA1c were performed only once, even when the results were positive. However, we believe that this does not affect the validity of our data, since GA and HbA1c due to their lifespan, unlike glycaemic tests, present good pre-analytical stability and less day-to-day variations during stress and illness.

## Conclusion

In conclusion, GA presented a good performance to detect newly diagnosed diabetes during COVID-19-related hospitalization. Admission value of GA of 19.0% may be useful cut-off to identify newly diagnosed diabetes. The cut-off had very high specificity but slightly low sensitivity. More studies that evaluate the clinical utility of GA on hospital admission and its association with complications are needed for a better understanding of the role of GA in hospitalized patients.

## Supporting information

**S1 Table. Dataset.** eGFR, estimated glomerular filtration rate by CKD-EPI Creatinine Equation; GA, glycated albumin; HbA1c, glycated haemoglobin.
(XLSX)

**S1 Dataset.**
(XLSX)

## Acknowledgments

We thank the participants of the HCPA Biobank. This research was conducted using the HCPA Biobank Resource.

## Author Contributions

**Conceptualization:** Fernando Chimela Chume, Joíza Lins Camargo.

**Data curation:** Fernando Chimela Chume, Luisa Gazzi Schiavenin, Eduarda Sgarioni.

**Formal analysis:** Fernando Chimela Chume, Joíza Lins Camargo.

**Funding acquisition:** Cristiane Bauermann Leitao, Joíza Lins Camargo.

**Investigation:** Fernando Chimela Chume, Priscila Aparecida Correa Freitas, Luisa Gazzi Schiavenin, Eduarda Sgarioni, Cristiane Bauermann Leitao, Joíza Lins Camargo.

**Methodology:** Fernando Chimela Chume, Priscila Aparecida Correa Freitas, Luisa Gazzi Schiavenin, Eduarda Sgarioni, Cristiane Bauermann Leitao, Joíza Lins Camargo.

**Project administration:** Fernando Chimela Chume, Luisa Gazzi Schiavenin, Joíza Lins Camargo.

**Resources:** Fernando Chimela Chume.

**Supervision:** Joíza Lins Camargo.

**Validation:** Fernando Chimela Chume, Priscila Aparecida Correa Freitas, Joíza Lins Camargo.

**Visualization:** Fernando Chimela Chume, Joíza Lins Camargo.

**Writing – original draft:** Fernando Chimela Chume.

**Writing – review & editing:** Fernando Chimela Chume, Priscila Aparecida Correa Freitas, Luisa Gazzi Schiavenin, Eduarda Sgarioni, Cristiane Bauermann Leitao, Joíza Lins Camargo.

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
