## [Decision Letter · Decision Letter 0]

3 Oct 2023

PONE-D-23-23999Glycated Albumin in the Detection of Diabetes During COVID-19 HospitalizationPLOS ONE Dear Dr. Camargo,

Thank you for submitting your manuscript to PLOS ONE. After careful consideration, we feel that it has merit but does not fully meet PLOS ONE’s publication criteria as it currently stands. Therefore, we invite you to submit a revised version of the manuscript that addresses the points raised during the review process.. Please, answer all reviewer's questions, in special one regarding the choice for the use of ROC curve of GA and not other glucose parameters (REVIEWER 1).

We look forward to receiving your revised manuscript.

Kind regards,

Fabio Vasconcellos Comim, MD,PhD

Academic Editor

PLOS ONE

Reviewers' comments:

Reviewer's Responses to Questions

**Comments to the Author**

1. Is the manuscript technically sound, and do the data support the conclusions?

Reviewer #1: Partly

Reviewer #2: Yes

2. Has the statistical analysis been performed appropriately and rigorously? 

Reviewer #1: N/A

Reviewer #2: I Don't Know

3. Have the authors made all data underlying the findings in their manuscript fully available?

Reviewer #1: No

Reviewer #2: Yes

4. Is the manuscript presented in an intelligible fashion and written in standard English?

Reviewer #1: Yes

Reviewer #2: Yes

5. Review Comments to the Author

Reviewer #1: This manscript by Fernando Chimela Chume investigated the role of GA in detecting newly diagnosed diabetes during COVID-19 hospitalization in a retrospective study. They enrolled 184 adults from the biobank and get the conclusion that GA higher than 19% is helpful for identify the newly diagnosed diabetes. The major concerns from this observation study is lack of the second indendpend test set for confirm the robust of this index.

some other concers need to be consider:

1) GA, HbA1c and the ratio of GA/HbA1c, even with glucose are also showed reliable diagnostic value in previous works (PMID: 28393586; PMID: 27386821). But the author only showed the ROC curve of the GA, it should be provide more comprehensive information for authors to compare with these known indexes to get the conclusion that GA is the best one.

2) Even this works carried out in the COVID-19 patients, I think the promising index for early diagnosis of newly diabetes is also important in those subjects without COVID-19 infection. The following question is how does the SARS-COV2 virus influent the level of GA? And What's the differential diagnosis value of GA for newly diabetes with other infectious disease?

3) The prospective study was carried out in COVID-19 patients from March 2020 to December 2020. It should be Original virus strain, have you compared to the newly virus strain with the original virus to explore the diagnostic value of GA of newly diabetes with COVID-19 infection?

4) It was reported the BMI correlates positively with HbA1c and negatively with GA. So the HbA1c may be more effective in obese and GA in nonobese individuals (PMID: 35783481; PMID: 17434227). In your study the average BMI of the subject is 29.4 (25.7, 34.5) that accros the overweiht and obesity group. How to consider the BMI in affecting the diagnostic value of GA?

Reviewer #2: 1- As they stated in their limitations section only once measurement for HbA1c and GA is the main limitation of the current study. Unfortunatelly, this has a big impact on the data validity.

2- GA could have fit to diagnose diabetes in different patient subsets, and COVID-19 is one of them.

3- It would be nice if the authors could provide their study group’s GA/ HbA1c ratio of their population and its possible clinical implications.

6. PLOS authors have the option to publish the peer review history of their article (what does this mean?). If published, this will include your full peer review and any attached files.

Reviewer #1: No

Reviewer #2: No

---

## [Author Response · Author response to Decision Letter 0]

15 Nov 2023

Dears Editor and Reviewers,

Thank you for allowing us the chance of re-submitted our manuscript. We believe that we have properly answered the Reviewer’s questions and/or suggestions. The text was amended accordingly. The evaluation and suggestions contributed to improve our manuscript.

The modifications are marked in the main text file.

Thank you for this observation. The manuscript has been amended accordingly.

Thank you, we have found that at our institution there are no ethical or legal restrictions on sharing an anonymized dataset. We added a table in our supplementary material, quoted in the manuscript (Page 23, Line 2) and update our Data Availability statement.

Thank you for this observation. We revised and amended accordingly.

ANSWERS TO REVIEWERS:

5. Review Comments to the Author

Reviewer #1: This manuscript by Fernando Chimela Chume investigated the role of GA in detecting newly diagnosed diabetes during COVID-19 hospitalization in a retrospective study. They enrolled 184 adults from the biobank and get the conclusion that GA higher than 19% is helpful for identify the newly diagnosed diabetes. The major concern from this observation study is lack of the second independent test set for confirm the robust of this index.

Answer: Thank you for this comment. We followed the Standard for Reporting Diagnostic Accuracy (STARD) initiative guideline to carry out this diagnostic accuracy study. It is recommended to evaluate the index test (i.e., GA) by the most suitable reference standard for the study setting. According to the literature (reference 22 in the manuscript file), HbA1c ≥6.5% is the choice during hospital admission. Therefore, we evaluated the performance of GA to identify patients presenting newly diagnosed diabetes by HbA1c ≥ 6.5% in our study. It is worth to mention that in this present study GA presented performance similar to those seen in outpatient’s studies (reference 27 in the manuscript file). Nevertheless, we agree with Reviewer 1 that comparison with other tests, as we often see in studies with outpatient participants, could provide us with additional information. However, in hospitalized patients tests such as fasting blood glucose and oral glucose tolerance tests are not feasible. Usually, the available test is random blood glucose (RBG) which is susceptible to pre-analytical interferences, including fasting, food intake, and acute illnesses such as COVID-19. We were able to have RBG results for our patients, and we built Receiver Operating Characteristic (ROC) curves to illustrate the performance of RBG and GA as index tests and HbA1c as reference test. The overall diagnostic accuracy for RBG and GA were similar in detecting newly diagnosed diabetes. Nonetheless, RBG is able to detect hyperglycaemia but it is unreliable in diagnosing diabetes in hospitalized patients due to above-mentioned interferences. To avoid misinterpretation, we chose not to show these results in the article.

 

Some other concerns need to be considered:

1) GA, HbA1c and the ratio of GA/HbA1c, even with glucose also showed reliable diagnostic value in previous works (PMID: 28393586; PMID: 27386821). But the author only showed the ROC curve of the GA, it should be providing more comprehensive information for authors to compare with these known indexes to get the conclusion that GA is the best one.

Thank you for this question. We agree that comparison with multiple indexes would give additional information. However, as we mentioned before, we used the most suitable reference standard available for the study setting, in our study was only HbA1c. Laboratorial tests that require fasting are not feasible during acute disease hospitalization.

The use of GA/HbA1c (as an index test) would be interesting, but once HbA1c was the reference test, it would give a mathematically inverted ROC curve due to the use of HbA1c as the denominator in the GA/HbA1c ratio. Below we show the ROC curve for GA/HbA1c ratio, but we chose not show these results in the article. Nevertheless, to present indirectly the mean amplitude of glycaemic excursion in our population we provided GA/HbA1c ratio results in Table 1 in the article file. 

  

2) Even this works carried out in the COVID-19 patients, I think the promising index for early diagnosis of newly diabetes is also important in those subjects without COVID-19 infection. The following question is how does the SARS-COV2 virus influent the level of GA? And What's the differential diagnosis value of GA for newly diabetes with other infectious disease?

Thank you for these questions. During the COVID-19 pandemic, several studies have reported that patients with diabetes mellitus (DM) have a higher risk of severe SARS-CoV-2 infection. Many reasons may explain this susceptibility, including the direct effect of hyperglycaemia in the immune system. However, this disease is characterized by a variety of clinical manifestations ranging from asymptomatic to severe symptoms that may contribute to glucose imbalance. In addition, the influence of genetic variations on clinical outcomes must be considered (References 1 to 4 in the manuscript file). The mechanism by how SARS-CoV-2 affects glucose and GA levels is unclear. The evaluation of the influence of SARS-CoV-2 on the blood GA level is out of the scope of the present study, but is clinically relevant, and it is our understanding that specific studies designed for this purpose are necessary. 

As far as we know, this is the first report to evaluate the GA performance in detecting newly diagnosed diabetes in acute ill (particularly COVID-19) individuals during hospital admission. Though data about GA performance in the diagnosis of diabetes in outpatients have been accumulating worldwide, there are no data on hospital admission subjects. We believe the test might be useful in other acute infectious diseases, but further studies are necessary for accurate conclusions about this issue. We amended the text to clarify these points (Page 16, paragraph 1, lines 13 – 19).

3) The prospective study was carried out in COVID-19 patients from March 2020 to December 2020. It should be Original virus strain, have you compared to the newly virus strain with the original virus to explore the diagnostic value of GA of newly diabetes with COVID-19 infection?

Thank you for this comment. We carried out our study retrospectively in samples available from a biobank in the first year of pandemic, before COVID-19 vaccines to reduce confounding factors, especially if SARS-CoV-2 influences on blood glucose and GA levels. However, we understand that additional studies in new scenarios with new virus strains are needed.

4) It was reported the BMI correlates positively with HbA1c and negatively with GA. So the HbA1c may be more effective in obese and GA in nonobese individuals (PMID: 35783481; PMID: 17434227). In your study the average BMI of the subject is 29.4 (25.7, 34.5) that across the overweight and obesity group. How to consider the BMI in affecting the diagnostic value of GA?

Thank you for this comment. We performed the correlations between HbA1c, GA, and BMI. The correlations were not significant (r = 0.020 and 0.029; p >0.100; results not shown in the article). These results are similar to those reported in our previous study (Ref 28 in the manuscript file).

Reviewer #2: 1- As they stated in their limitations section only once measurement for HbA1c and GA is the main limitation of the current study. Unfortunately, this has a big impact on the data validity.

Thank you for this comment. This is a limitation of all cross-sectional analysis where longitudinal analyses are not performed. However, we believe that this does not affect the validity of our data, since GA and HbA1c, unlike glycaemic tests, present good pre-analytical stability and less day-to-day variations during stress and illness due to their lifespan. We amended the text for clarity accordingly (Page 16, paragraph 3, lines 30 – 34).

2- GA could have fit to diagnose diabetes in different patient subsets, and COVID-19 is one of them.

Thank you for this comment. We agree with the reviewer that GA can detect diabetes in different subsets of individuals. As we mentioned previously, data on other subsets are needed for accurate conclusions and adequate clinical use of GA. We believe that our study is the first to assess the performance of GA on admission of hospitalized adults, particularly with COVID-19, and our data offers to the literature more evidence that this test may be useful in different settings to diabetes diagnosis.

3- It would be nice if the authors could provide their study group’s GA/ HbA1c ratio of their population and its possible clinical implications.

Thank you for this comment. We understand that evaluating GA/HbA1c ratio clinical implications is beyond the scope of this study. The GA/HbA1c ratio is related to the mean amplitude of glycaemic excursion and is suggested as short-term glycaemic control marker. Nevertheless, as mentioned above, to present indirectly the mean amplitude of glycaemic excursion in our population we provided GA/HbA1c ratio results in Table 1 in the article file. 

Finally, we would like to express our appreciation to you and the reviewers for suggesting how to improve our paper. 

Thank you very much,

Yours sincerely,

Joíza Lins Camargo, PhD

Experimental Research Centre

Hospital de Clínicas de Porto Alegre

Rua Ramiro Barcellos, 2350 – 1o andar

Porto Alegre, Brazil 90035-006

e-mail: jcamargo@hcpa.edu.br; Phone +55 51 33598852

---

## [Decision Letter · Decision Letter 1]

11 Dec 2023

PONE-D-23-23999R1Glycated Albumin in the Detection of Diabetes During COVID-19 HospitalizationPLOS ONE

Dear Dr. Camargo,

Thank you for submitting your manuscript to PLOS ONE. After careful consideration, we feel that it has merit but does not fully meet PLOS ONE’s publication criteria as it currently stands. Therefore, we invite you to submit a revised version of the manuscript that addresses the points raised during the review process.

We look forward to receiving your revised manuscript.

Kind regards,

Fabio Vasconcellos Comim, MD,PhD

Academic Editor

PLOS ONE

Journal Requirements:

Reviewer's Responses to Questions

**Comments to the Author**

1. If the authors have adequately addressed your comments raised in a previous round of review and you feel that this manuscript is now acceptable for publication, you may indicate that here to bypass the “Comments to the Author” section, enter your conflict of interest statement in the “Confidential to Editor” section, and submit your "Accept" recommendation.

Reviewer #3: (No Response)

2. Is the manuscript technically sound, and do the data support the conclusions?

Reviewer #3: Yes

3. Has the statistical analysis been performed appropriately and rigorously? 

Reviewer #3: Yes

4. Have the authors made all data underlying the findings in their manuscript fully available?

Reviewer #3: Yes

5. Is the manuscript presented in an intelligible fashion and written in standard English?

Reviewer #3: Yes

6. Review Comments to the Author

Reviewer #3: I read with interest the paper of Camargo et al regarding the role of glycated albumin to detect diabetes mellitus in patients hospitalized for COVID-19 infection.

The paper is interesting, however some drawbacks need to clarified. The main point is the lack of an independent test to confirm the GA cut-off used to diagnose diabetes mellitus.

Corticosteroid therapy should be considered, especially if patients had a relapse of COVID-19, since it may have affected the GA results.

Elevated levels of GA have been correlated to COVID-19 severity (high CPR levels) (J. Clin. Med. 2022, 11(9), 2327). Only 15 of 38 patients with elevated GA have also high HbA1 levels. Do the authors have any data regarding patients’ follow-up in term of in hospital stay and/or death? Is there a stronger correlation between GA levels and outcome rather than HbA1?

7. PLOS authors have the option to publish the peer review history of their article (what does this mean?). If published, this will include your full peer review and any attached files.

Reviewer #3: No

---

## [Author Response · Author response to Decision Letter 1]

2 Jan 2024

Dears Editor and Reviewers,

Thank you for allowing us the chance of re-submitted our manuscript. We believe that we have properly answered the Reviewer’s questions and/or suggestions. The text was amended accordingly. The evaluation and suggestions contributed to improve our manuscript.

The modifications are marked in the main text file.

6. Review Comments to the Author

Reviewer #3: I read with interest the paper of Camargo et al regarding the role of glycated albumin to detect diabetes mellitus in patients hospitalized for COVID-19 infection.

The paper is interesting, however some drawbacks need to clarified. The main point is the lack of an independent test to confirm the GA cut-off used to diagnose diabetes mellitus.

Thank you for this comment. Reviewer #1 had already raised this concern. As we explained in Reviewer #1's answer, we followed the Standard for Reporting Diagnostic Accuracy (STARD) initiative guideline to carry out this diagnostic accuracy study. It is recommended to evaluate the index test (i.e., GA) by the most suitable reference standard for the study setting. According to the literature (reference 22 in the manuscript file), HbA1c ≥6.5% is the choice during hospital admission. Therefore, HbA1c was the independent test and we evaluated the performance of GA to identify patients presenting newly diagnosed diabetes by HbA1c ≥ 6.5% in our study.

Corticosteroid therapy should be considered, especially if patients had a relapse of COVID-19, since it may have affected the GA results.

Thank you for this observation. Revising our data, we found that all analyses were carried out in blood samples collected at the beginning of hospitalization before starting in-hospital treatment, and no individual had a relapse of COVID-19 at the time of the study. Nevertheless, it is our understanding, that there was not enough time for corticosteroid therapy to affect GA results, or even HbA1c results. Since, the length of hospital stay of our participants had a median of 9 (interquartile range: 4, 16) days, GA test is limited to a three-week mean glycemia (the average albumin turnover). On the other hand, HbA1c is dependent on haemoglobin, which has a much longer turnover period. We amended the text for clarity accordingly (Page 8, Lines 12 and 13)

Elevated levels of GA have been correlated to COVID-19 severity (high CPR levels) (J. Clin. Med. 2022, 11(9), 2327). Only 15 of 38 patients with elevated GA have also high HbA1 levels. Do the authors have any data regarding patients’ follow-up in term of in hospital stay and/or death? Is there a stronger correlation between GA levels and outcome rather than HbA1?

Thank you for this comment. Although is clinically relevant, evaluating the association between GA and clinical outcomes such as length of hospital stay or death is beyond the scope of the present study, besides would require a long-term study for accurate analyses and interpretation. The present study was designed to perform transversal analysis to evaluate the diagnostic accuracy of GA in identifying unknown diabetes on hospital admission and followed the Standard for Reporting Diagnostic Accuracy (STARD) initiative guideline. However, we understand its importance, and it can be addressed in future studies.

Finally, we would like to express our appreciation to you and the reviewers for suggesting how to improve our paper. 

Thank you very much,

Yours sincerely,

Joíza Lins Camargo, PhD

Experimental Research Centre

Hospital de Clínicas de Porto Alegre

Rua Ramiro Barcellos, 2350 – 1o andar

Porto Alegre, Brazil 90035-006

e-mail: jcamargo@hcpa.edu.br; Phone +55 51 33598852

---

## [Decision Letter · Decision Letter 2]

16 Jan 2024

Glycated Albumin in the Detection of Diabetes During COVID-19 Hospitalization

PONE-D-23-23999R2

Dear Dr. Camargo,

We’re pleased to inform you that your manuscript has been judged scientifically suitable for publication and will be formally accepted for publication once it meets all outstanding technical requirements.

Kind regards,

Fabio Vasconcellos Comim, MD,PhD

Academic Editor

PLOS ONE

Reviewer #3: All comments have been addressed

2. Is the manuscript technically sound, and do the data support the conclusions?

Reviewer #3: Yes

3. Has the statistical analysis been performed appropriately and rigorously? 

Reviewer #3: Yes

4. Have the authors made all data underlying the findings in their manuscript fully available?

Reviewer #3: Yes

5. Is the manuscript presented in an intelligible fashion and written in standard English?

Reviewer #3: Yes

6. Review Comments to the Author

Reviewer #3: Authors replayed to all my quires.

The paper is well written and authors properly explained the findings.

7. PLOS authors have the option to publish the peer review history of their article (what does this mean?). If published, this will include your full peer review and any attached files.

Reviewer #3: No

---

## [Editor Report · Acceptance letter]

4 Mar 2024

PONE-D-23-23999R2 

PLOS ONE

Dear Dr. Camargo, 

I'm pleased to inform you that your manuscript has been deemed suitable for publication in PLOS ONE. Congratulations! Your manuscript is now being handed over to our production team.

Kind regards, 

on behalf of

Prof Fabio Vasconcellos Comim 

Academic Editor

PLOS ONE